# Bioactive Components and Their Activities from Different Parts of *Citrus aurantifolia* (Christm.) Swingle for Food Development

**DOI:** 10.3390/foods12102036

**Published:** 2023-05-17

**Authors:** Nastiti Nur Indriyani, Jamaludin Al Anshori, Nandang Permadi, Sarifah Nurjanah, Euis Julaeha

**Affiliations:** 1Department of Chemistry, Faculty of Mathematics and Natural Sciences, Universitas Padjadjaran, Jatinangor 45363, Indonesia; nastiti21002@mail.unpad.ac.id (N.N.I.); jamaludin.al.anshori@unpad.ac.id (J.A.A.); 2Doctorate Program in Biotechnology, Graduate School, Universitas Padjadjaran, Bandung 40132, Indonesia; nandang17001@mail.unpad.ac.id; 3Department of Agricultural Engineering, Faculty of Agricultural Industrial Technology, Universitas Padjadjaran, Jatinangor 45363, Indonesia; sarifah@unpad.ac.id

**Keywords:** bioactivity, human health, phytochemical, *Citrus aurantifolia*, extraction

## Abstract

*Citrus aurantifolia* is part of the Rutaceae family and belongs to the genus Citrus. It is widely used in food, the chemical industry, and pharmaceuticals because it has a unique flavor and odor. It is nutrient-rich and is beneficial as an antibacterial, anticancer, antioxidant, anti-inflammatory, and insecticide. Secondary metabolites present in *C. aurantifolia* are what give rise to biological action. Flavonoids, terpenoids, phenolics, limonoids, alkaloids, and essential oils are among the secondary metabolites/phytochemicals discovered in *C. aurantifolia*. Every portion of the plant’s *C. aurantifolia* has a different composition of secondary metabolites. Environmental conditions such as light and temperature affect the oxidative stability of the secondary metabolites from *C. aurantifolia*. The oxidative stability has been increased by using microencapsulation. The advantages of microencapsulation are control of the release, solubilization, and protection of the bioactive component. Therefore, the chemical makeup and biological functions of the various plant components of *C. aurantifolia* must be investigated. The aim of this review is to discuss the bioactive components of *C. aurantifolia* such as essential oils, flavonoids, terpenoids, phenolic, limonoids, and alkaloids obtained from different parts of the plants and their biological activities such as being antibacterial, antioxidant, anticancer, an insecticide, and anti-inflammatory. In addition, various extraction techniques of the compounds out of different parts of the plant matrix as well as the microencapsulation of the bioactive components in food are also provided.

## 1. Introduction

Citrus is a commonly consumed fruit because it contains many beneficial nutrients [1]. One of the citrus species that is widely used is *C. aurantifolia*. The species *C. aurantifolia* belongs to the Rutaceae family, which consists of 150 genera and 900 species [2]. This plant has been widely cultivated throughout the world [3]. *C. aurantifolia* is a small herbaceous plant with a distinctive odor. The fruit is slightly round in shape with a tapered end. In addition, the fruit has a very sour, juicy taste and a strong aroma. *C. aurantifolia* itself is widely used as a raw material for cosmetics, food flavoring, flavor enhancers in beverages, and as an ingredient in traditional medicine [4].

*C. aurantifolia* has various biological activities [5]. Several studies have found *C. aurantifolia* to have biological activities such as being an insecticide, larvicide, and repellent [6]; antioxidant, anticancer, and antimicrobial [7,8]; antiseptic, antiviral, antifungal, astringent, anticholesterol, diuretic, appetite stimulant, constipation remedy [9], anti-inflammatory, and analgesic [10]. These various biological activities are due to the content of secondary metabolites found in *C. aurantifolia*. Secondary metabolites found in *C. aurantifolia* include alkaloids, coumarins, flavonoids [11], carotenoids, phenolics, terpenes, limonoids [12], and essential oils [13]. The content of these secondary metabolites can be influenced by various factors, such as physicochemical properties, soil composition, sun exposure, geographical coordinates [14], and the part of the plant used [15].

Bioactive components are generally prone to degradation, either during storage or food processing. This can be caused by physical, chemical, or enzymatic changes that are unstable, causing degradation and transformation. As a result, the bioactivity of these bioactive compounds can be lost. It can be difficult to understand the mechanism behind the degradation of bioactive compounds due to their complexity. The microencapsulation method can be used as a tool to protect natural extracts and bioactive components. Microencapsulation is a method used to carry out, protect, and transport active chemicals (core material) in their precise region of action by entrapping them in a surrounding material (coating agent, encapsulating agent, or wall material) [16]. With microencapsulation, factors such as oxygen, heat, light, and humidity that can affect the stability of bioactives can be prevented [17]. The advantages of microencapsulation, which include handling convenience and control of the release and solubilization of active ingredients, open up a huge space for the progress of food science and processing. Microcapsules can be used successfully in a variety of food matrices, including meat products, dairy products, cereals, and fruits, as well as in their derivatives, to develop functional food products, reduce fat, improve sensory perception, preserve food, and other applications [18]. Therecognition of *C. aurantifolia* is because of its flavor and odor, as well as the nutritional value provided by the plant’s bioactive components. Because every part of the plant has its unique bioactive components and biological activities, it is necessary to investigate these substances more thoroughly. There is currently no review of the chemical makeup of different plant sections, biological activity, or application of *C. aurantifolia* as food-grade microcapsules. Therefore, this review article discusses the bioactivity of compounds from *C. aurantifolia* plant parts, including leaves, fruit, peels, roots, and seeds, as well as their bioactivity as an anticancer, antioxidant, anti-inflammatory, and other bioactivities to maximize health benefits—specifically, those that were taken from various *C. aurantifolia* parts using various extraction techniques. The microencapsulation methods for increasing the stability of bioactive components from *C. aurantifolia* were investigated. For better advancement in the areas of research, application, or industrial production, it is anticipated that this review will be able to increase scientific understanding and further utilization of *C. aurantifolia*.

## 2. Taxonomy

*C. aurantifolia* has spread worldwide, from Southeast Asia to Brazil [19]. *C. aurantifolia* is commonly known as the Key lime and can grow widely in subtropical and tropical regions [20]. This plant can reach a height of 3–6 m, with numerous branches and irregular thorns covered in smooth brown-to-gray bark. The leaves are small and oval-shaped, with winged petioles. Crushed leaves have a characteristic odor and a strong nutty taste. The flowers appear on the branching leaves and are white, small in size, and have a fragrant smell, numbering four to five. The fruit is round to ovular with a diameter of 3–5 cm, is green to yellow with thin peel, and contains a lot of essential oils. The fruit contains several white seeds approximately 1 cm long [13,19,21]. *C. aurantifolia* in some areas is known by names such as lime (English), jeruk nipis (Indonesia), limonene (German), lime (Malaysia), citronnier (France), lima acida (Italy), oman lime (Oman), zhi qiao (China), Kagzi-nimu (India), chah ta (Vietnam), manao (Thailand), limao galego (Portuguese, lima agria (Spain), jeruk pecel (Java) and limoo (Persia) [11]. The taxonomy of *C. aurantifolia* is as follows:
KingdomPlantaeSubkingdomTracheobiontaSuperdivisionSpermatophytaDivisionMagnoliophytaClassMagnoliopsidaSubclassRosidaeOrderSapindalesFamilyRutaceaeGenusCitrusSpecies*Citrus aurantifolia* [11]

## 3. Extraction

The use of bioactive compounds in various fields, such as pharmaceuticals, the chemical industry, and food, has led to the development of suitable and standardized methods for extracting bioactive compounds. There are various extraction methods to obtain bioactive compounds. However, no single method can be used as a standard for extracting bioactive compounds from their native matrices. The efficiency of using conventional and non-conventional extraction methods is influenced by several factors, such as plant matrices, bioactive compounds in plants, and scientific expertise [22]. In addition, the extraction results of these bioactive compounds are strongly influenced by the conditions of the experiment [23,24]. In general, the extraction efficiency of bioactive compounds from plants depends on the type of solvent, extraction time, pH value, and temperature [25]. The extraction at elevated temperature increases the solubility and dilution of the substances. However, excessively high temperatures can result in the loss of solvents, which can extract unwanted impurities and induce the destruction of thermolabile components. With an increase in extraction time over a specific time period, the extraction efficiency rises. Once the solute has reached equilibrium both inside and outside of the solid substance, adding more time will not have any effect on the extraction. The extraction yield increases with the solvent-to-solid ratio; however, a high solvent-to-solid ratio will result in surplus extraction solvent and take a long time to concentrate [26].

The composition and yield of the extraction are also affected by the extraction method used, as each method has a different effect on the sample. Conventional methods such as hydrodistillation, maceration, and soxhlation have been used for a long time for the extraction of bioactive compounds [27]. In most instances, the standard and reference method for solid–liquid extraction has been the Soxhlet apparatus, which is still widely used in laboratories and throughout the industry. However, there are numerous drawbacks to Soxhlet extraction, including the need for long operating times (several hours), large solvent volumes, the need for concentration and evaporation at the end of the extraction, and insufficiency for thermolabile analytes [28].

To extract the compounds, the solvent must be carefully chosen based on factors including selectivity, solubility, cost, and safety should be taken into account. The law of resemblance and intermiscibility (like dissolves like) states that solvents having polarity values close to the polarity of the solute would likely perform better, and vice versa [26]. The solvents for extraction that are commonly used are acetone, methanol, ethanol, ethyl acetate, and propanol [29]. Solvents with high polarities, such as methanol, can dissolve compounds such as phenol. However, solvents such as methanol were identified as toxic, so many then used ethanol as an extraction solvent [30]. 

With the development of technology, advanced techniques have been found that can be used for extraction [27], one of which is supercritical-CO_2_ extraction. This extraction method can reduce the amount of organic solvent used in the extraction process, making it more environmentally friendly. However, this method requires high costs, sophisticated equipment, and special expertise from users [31]. Other modern extraction methods, such as ultrasound-assisted extraction (UAE), which is based on cavitation and breaks down cell walls, are also available. This process can cut expenses, save time, and yield better outcomes [32]. Several schemes for extracting secondary metabolites can be seen in Figure 1 [33,34,35,36].

A comparison of the extraction methods used to extract secondary metabolites of *C. aurantifolia* is provided in Table 1. The data revealed that such factors as the extraction method, plant parts, and solvents determined the yield of the extracted compounds [37].

Therefore, it is necessary to understand the structure and composition of the active substances being extracted, as well as their own matrix, to obtain the most appropriate extraction method. Such green chemistry principles are applied to achieve a low environmental impact and a sustainable process, employing technologies with high energy efficiency and preferably from renewable sources [45].

## 4. Chemical Components in Parts of *C. aurantifolia*

Plants produce chemical components known as secondary metabolites. These metabolites are not directly involved in the growth process. However, this metabolite acts as a deterrent against microbial and insect attacks [46]. Secondary metabolites can be divided into volatile components such as essential oils and non-volatile components such as flavonoids, terpenoids, phenolics, limonoids, and alkaloids.

The bioactive components present in *C. aurantifolia* can be identified and characterized from plant parts such as stems, leaves, flowers, seeds, and fruit [47]. Citrus contains a variety of bioactive components. Citrus fruit peels that have not been widely used contain secondary metabolites with high antioxidant activity when compared to other parts. In addition, citrus peels also contain molasses, pectin, and limonene [15].

The number of compounds extracted from the peel of *C. aurantifolia* using ethyl acetate, chloroform, and *n*-hexane were 28, 27, and 24 compounds, respectively. The majority of the 79 compounds were active compounds such as α-tocopherol, d-limonene, and phytol [7]. Some of the bioactive components present in the parts of *C. aurantifolia*, which were identified phytochemically, can be seen in Table 2.

Based on Table 2, it can be seen that the phytochemical screening of plant parts of *C. aurantifolia* produced different bioactive component test results. Identified essential oils can be obtained from the leaves and fruit peels [54]. Flavonoids can be found in all parts of the *C. aurantifolia* plant, starting from the leaves, fruit, fruit peel, seeds, stems, roots, and bark [48]. Likewise, the terpenoids, phenolics, saponins, and alkaloids in *C. aurantifolia* were found in several parts of the plant. The bioactive components of *C. aurantifolia* are not found in all parts of the plant. Each of the bioactive components derived from parts of *C. aurantifolia*, such as essential oils, flavonoids, terpenoids, phenolics, limonoids, and alkaloids, will be explained further in the next sections.

### 4.1. Essential Oil

*C. aurantifolia* essential oil is an oily liquid obtained from the lime plant. This essential oil can be obtained by hydrodistillation, steam distillation, maceration, soxhlation, and cold pressing methods. However, the extraction method by steam distillation is the most widely used extraction method [55]. As much as 93% of essential oils are extracted commercially by traditional methods such as distillation, and the remaining 7% by other methods [56]. This method is relatively simple and considered an economical method for obtaining essential oils (with a better yield of about 0.21%) when compared to other methods such as cold pressing, with a yield of around 0.05% [56]. The essential oil and its constituents are extracted and isolated from either citrus peel, leaves, or flower [57].

The essential oil of the peel of *C. aurantifolia* consists of 75% terpenes, 12% oxygenated compounds, and 3% sesquiterpenes. Monoterpenes (such as d-limonene and γ-terpinene), sesquiterpenes, hydrocarbons (and their oxygenated derivatives such as geranial, nonanal, neryl, and linalool), including aldehydes, ketones, acids, alcohols, and esters, are among the volatile constituents. Volatile compounds are further categorized as alcohols, ethers, aldehydes, ketones, esters, amines, amides, phenols, heterocyclics, and especially terpenes [27]. Non-volatile compounds consist of fatty acids, long-chain hydrocarbons, sterols, waxes, and limonoids [58]. According to research by Puspita et al. (2020) [59] and Spadaro et al. (2012) [60], the main components of *C. aurantifolia* essential oil are d-limonene (35.98%), β-pinene (9.02%), α-terpineol (8.12%), and citral (7.49%). Other components contained in *C. aurantifolia* essential oil are linalool, linalyl acetate, geraniol, citral, β-pinene, α-terpineol, felandrene, sesquiphellandrene, citronellol, neryl acetate, fencone, farnesene, and geranyl acetate. The structure of the compounds contained in *C. aurantifolia* essential oil can be seen in Figure 2 [61,62].

*C. aurantifolia* essential oil has several biological activities, including as an anticancer, antioxidant, antiulcer, antimicrobial, anti-inflammatory, antityphoid, hypolipidemic, and hepatoprotective. It also has strong antibacterial and antifungal activity. This causes *C. aurantifolia* essential oil to be used as an important component in skin care products [61].

### 4.2. Flavonoids

Flavonoids are secondary metabolites derived from 2-phenyl-benzyl-γ-pyrone that are found in most plants. A total of more than 9000 components of the flavonoid group have been studied [63]. Flavonoids in Citrus have a wide range of biological activities such as free radical scavengers (antioxidants), modulating enzyme activity, inhibiting cell proliferation, and being antibiotic, hypoallergenic, antiulcer, antidiarrheal, and anti-inflammatory [14]. The types of flavonoids found in *C. aurantifolia* extract are apigenin, rutin, kaempferol, quercetin, and nobiletin [14]. In addition, *C. aurantifolia* also contains flavonoids such as eriocitrin and hesperidin, which have antioxidant activity [64]. The highest hesperidin levels were found in fruit contents (58.43 µg/g dry weight) and fruit peel (32.49 µg/g dry weight). However, hesperidin is not found in the leaves [65]. Hesperidin in lime peel had a 5.8 to 11 times increase in antioxidant activity compared to unfermented lime peel. The high antioxidant activity of fermented lime peel may be related to the biotransformation of hesperidin to hydroxy flavanone and aglycone, given that other low-molecular-weight phenolic compounds are produced [66].

When compared to seeds, juice, and fruit, the fruit peel contains the most flavonoids in *C. aurantifolia*, including the mesocarp, epicarp, and layer segments [67]. Flavonoid content ranges from 41.38–64.2 μg of QE/mg of methanol dry extract [11]. Flavonoids found in the peel of *C. aurantifolia* include apigenin, rutin, quercetin, and kaempferol [13]. The distribution of quercetin is slightly different from that of the hesperidin compound, where the highest level of quercetin was found in the fruit peel, which was 112.47 µg/g dry weight, and the lowest level was found in the fruit content, which was 89.4947 µg/g dry weight. In the fruit leaves, there was quercetin with a level of 92.71 µg/g dry weight [65], while the total flavonoids in *C. aurantifolia* juice obtained levels of 29.5 QE mg/L [68]. The methanolic extract of *C. aurantifolia* fruit was also reported to contain eriocitrin, hesperidin, naringin, and naringenin, which belong to the class of flavonoids [50].

Research by Herawati et al. (2020) [63] showed that flavonoid assay of the ethanolic extract of *C. aurantifolia* leaves using a UV-Vis spectrophotometer showed a flavonoid level of 1.12%. Similarly, Swandiny et al. (2021) [4] reported a subtle different of flavonoid content between the extract macerated with 96% methanol (1.54%) and with 70% methanol (1.485%). Further HPLC analysis reported by Loizzo et al. (2012) [14] confirmed the typical flavonoids of methanolic extract of the leaves were rutin, apigenin, quercetin, kaempferol, nobiletin, and tangeretin. Meanwhile, the flavonoid content in *C. aurantifolia* seeds was 0.00842 ± 0.01% [53], lower than those in the roots (0.64 ± 0.40%), stems (0.33 ± 0.01%), bark (0.42 ± 0.01%), and leaves (0.06 ± 0.07%) reported by [69].

The use of different solvents in the extraction process also produces different levels of flavonoids. Research results from Karatoprak et al. (2021) [41] showed that the dry extract of *C. aurantifolia* fruit using ethanol solvent had the highest total flavonoid content, with a value of 7.83 ± 2.66 mg CA/g extract, when compared to using distilled water and methanol solvent, which produced total flavonoid levels of 6.52 ± 0.77 and 6.27 ± 0.39 mg CA/g extract. The structure of the flavonoids in *C. aurantifolia* can be seen in Figure 3 [13,70].

### 4.3. Terpenoids

Terpenoids are components with low molecular weights that have volatile properties and are usually found as components that make up essential oils. Terpenoids can act as an antibacterial. The mechanism of action is to break down the membrane because it has lipophilic properties. In addition, terpenoids can make the cytoplasmic membrane their main target because of their hydrophobic nature. Mirnawati et al. (2021) [71] discovered that the terpenoid content in the ethanolic extract of *C. aurantifolia* bark contained 34.81% β-pinene and 20.15% d-limonene. Monoterpenes contain the chemical components β-pinene and d-limonene, both of which have ten carbons. This compound is reported to have several biological activities, such as antibacterial, antiseptic, and anticancer properties. The terpenoids present in *C. aurantifolia* can be divided into monoterpenes, alcoholic terpenes, aldehyde terpenes, ketone terpenes, and ester terpenes [54]. The structure of the terpenoid compounds found in *C. aurantifolia* can be seen in Figure 4 [54].

### 4.4. Phenolic

Phenolic compounds have an important role in the plant defense system against various diseases caused by fungi, viruses, and bacteria [72]. Total phenolic content matched the antioxidant activity [73]. The levels of phenolic compounds in *C. aurantifolia* fruit can be influenced by the level of maturity of the fruit, where the highest phenolic content is found in immature fruit [74]. It is also influenced by the solvent used in the extraction process. Karatoprak et al. (2021) [41] mentioned that the ethanolic extract has a higher total phenolic content when compared to other solvents such as distilled water and methanol. Likewise, this is the case with the concentration of solvent used. According to Swandiny et al. (2021) [4], the total phenolic content in *C. aurantifolia* leaf extract when extracted using 96% alcohol produced a higher total phenolic content when compared to 70% alcohol.

The pulp of *C. aurantifolia* has four phenolic acids in the form of gallic acid (26.85 g/g) which is the largest component, tannic acid (14.27 g/g), and ferulic and coumaric acids which are reported as traces [72]. The structure of the phenolic compounds from *C. aurantifolia* can be seen in Figure 5 [72,75].

Al Namani et al. (2018) [11] reported that the total phenolic content of *C. aurantifolia* leaf extract, which was determined using the Folin Ciocalteu Reagent (FCR) and expressed in Gallic Acid Equivalent (GAE)/mg dry extract, ranged from 96.55–322.57 μg of GAE/mg dry extract. *C. aurantifolia* essential oil also contains phenolic compounds at levels of 285.80 GAE/mL which have antiseptic activity [76]. A comparison of total phenolic content in *C. aurantifolia* plant parts can be seen in Table 3.

### 4.5. Limonoids

Limonoids are one of the chemical components found in citrus. There are two classes of limonoids in Citrus i.e., aglycones and glycosides. Aglycones can be divided into neutral dilactones, acidic mono-lactones, and dicarboxylic acids. Generally, limonoid aglycones have a bitter taste and are insoluble in water. This compound is what ultimately gives oranges a bitter taste [77]. The most important constituents are glycosides called limonin and nomilin [78]. Limonoids belong to a unique class of oxygenated tetracyclic triterpenoids. These compounds have biological activities such as being anticancer, antioxidant, antibacterial, larvicidal, antimalarial, antiviral [79], hypoallergenic, anti-inflammatory, antiproliferative, antimutagenic, anticarcinogenic [80], antitumor, antiobesity, and antihyperglycemic [81]. Limonoids can be found in various parts of citrus plants, such as the seeds, fruit, bark, and roots [80]. There are more than 60 types of limonoids that have been isolated and characterized from the Citrus [82].

Castillo-Herrera et al. (2015) [44] used the supercritical-CO_2_ extraction method to obtain limonoids from *C. aurantifolia* seeds, using methanol and acetone as solvents. The limonoid content in methanol solvent was 1.65 mg/g, while in acetone solvent it was 0.779 mg/g. The limonoid derivative compounds found in *C. aurantifolia* can be seen in Figure 6 [81].

### 4.6. Alkaloids

The main alkaloids found in the genus Citrus are synephrine, tyramine, and octopamine [83]. The structure of these compounds can be seen in Figure 7 [83]. One of the developments in the alkaloid analysis method that has been developed is the use of the HPLC-DAD-ESI/MS method, which can work quickly and accurately. This method can simultaneously measure the bioactive constituents in different fruits, such as those in the Citrus genus. Based on measurements made by He et al. (2011) [84], it is known that the total alkaloids in Citrus peel are higher when compared to Citrus pulp. In the species *C. aurantifolia* itself, the results of the phytochemical screening of the ethanolic extract of *C. aurantifolia* leaves were positive for alkaloids [85]. Dried *C. aurantifolia* fruit was reported to contain alkaloids of 0.33 ± 0.11 mg/100 g [46]. Citrus alkaloids and glycosides have biological properties such as anticancer activity and can be used as drug supplements [86].

## 5. Biological Activities

Secondary metabolite compounds derived from plants such as *C. aurantifolia* have benefits for human health. This secondary metabolite has a wide range of biological activities, including antibacterial, anticancer, antioxidant, insecticide, and anti-inflammatory properties. Each of the biological activities derived from parts of *C. aurantifolia* will be explained further in the next sections.

### 5.1. Antibacterial

According to most researchers’ findings, *C. aurantifolia* extract is potentially employed as a pure or a mixture of herbs for treating diseases and infections [87]. In particular, the antibacterial activity is because of its proteolytic and lipolytic properties [88] which are affected by their bioactive substances such as phenols, flavonoids, and hydrogen peroxide. Further investigation showed that the antibacterial activity mechanism of *C. aurantifolia* caused by phenolic compounds and their derivatives was through the denaturation process of bacterial cell proteins. One of the phenolic derivatives, i.e., chavicol, exhibited an anti-bactericidal activity five times stronger than phenol [76]. According to Galovičová et al. (2022) [6], the antibacterial activity of *C. aurantifolia* was better on Gram-negative bacteria than on some Gram-positive bacteria.

The ethanolic extract of *C. aurantifolia* leaves exhibited higher antibacterial activity than the aqueous one against *Klebsiella* bacteria and *Escherichia coli*, though the isolates showed distinct susceptibility to one another [89]. It is possible that the higher activity of the ethanolic extract can be attributed to the better solubility of the secondary metabolites than the aqueous ones, thus leading to better efficacy. However, out of bioactive components, the antibacterial activity of plants was also dependent linearly on their concentration [90].

Some of the antibacterial activity of *C. aurantifolia* plant parts are described in Table 4.

### 5.2. Antioxidant

The antioxidant activity of *C. aurantifolia* was reported due to the main flavonoids such as hesperidin, nobiletin, and tangeretin, which were accumulated mostly in the fruit peels and leaves. Thus, it was potentially applied as a good source of natural antioxidants. Further molecular docking study revealed the inhibition mechanism of the compounds towards the main protease (Mpro) and spike (S) glycoprotein enzymes [101].

On the other hand, a recognized in vitro assay of radical scavengers, DPPH, ABTS, FRAP, and beta-carotene bleaching tests were employed by Loizzo et al. (2012) [14] to assess the antioxidant activity of *C. aurantifolia*. A concentration–response relationship between the methanolic extract of peel and leaf was observed. The reducing ability showed a similar trend, with values ranging from 112.1 to 146.0 µmol L^−1^ Fe(II) g^−1^. Accordingly, Karatoprak et al. (2021) [41] also found 70% better antioxidant performance in the methanolic extract than the ethanolic ones which was vice versa for their cytotoxicity effect. In addition, both extracts demonstrated a significant hypoglycemic effect. Further investigation related to the citrus maturity and its storage system affecting the antioxidant activity was conducted. Likewise, the FRAP activity, the inhibition of DPPH free radicals by the citrus juices obtained from immature fruits commonly showed higher activity than the mature one [74,102]. According to Azman et al. (2019) [103], the antioxidant activity of frozen citrus peels exhibited higher antioxidant activity than the fresh peels. This was because of the higher phenolic content in the frozen peel. Based on the DPPH assay, the antioxidant activity of *C. aurantifolia* was commonly higher than the lemons, i.e., 17.21 ± 1.6. The comparison of the antioxidant activity of *C. aurantifolia* plant parts is presented in Table 5.

### 5.3. Anticancer

For several years, research on citrus has shown that it can prevent many diseases, one of which is cancer [109]. In vitro and in vivo studies have shown that the compounds found in citrus have anticancer activity. Several compounds that are suspected of having anticancer activity are found in citrus: limonoids, flavonoids, essential oils, coumarins, vitamins, and fatty acids [110].

Quercetin compounds, which are mainly found on the peel of *C. aurantifolia*, also have potential anticancer agents. Research results from Kenyori et al. (2022) [111] in silico that quercetin in the peel of *C. aurantifolia* has the potential as an anti-breast cancer agent with a binding affinity of −8.2 kcal/mol. These values indicate that quercetin has a stronger bond with its receptor than the original ligand (DRO with IUPAC name 1-(2-{[3S)-3-(aminomethyl)-3,4-dihydroxyisoquinolin-2(1H)yl]carbonyl}phenyl-4-chloro-5-methyl-N,N-diphenyl-1H-pyrazole-3-carboxamide). Quercetin also has interactions with amino acid residues, which can prove that this compound has the potential to be used as an anti-breast cancer drug. Another in silico analysis of β-pinene and limonene compounds in *C. aurantifolia* essential oil revealed anticancer activity. In the regulation of apoptosis, β-pinene, and limonene compounds have a lower affinity for the native ligand present in the caspase-8 protein [112]. The effect of *C. aurantifolia* juice extract on the spontaneous proliferation of a breast carcinoma cell line (MDA-MB-453) and lymphoblastoid B cells (RPMI-8866) was also investigated in vitro. The extract had no significant effect on MDA-MB-453 cells. However, when used at 125, 250, and 500 g/mL concentrations, the extracts showed significant inhibition of the spontaneous proliferation of the rpmi-8866 cell line [113].

Caryophyllene compounds also have antibacterial and anticancer properties. These compounds have a selective cytotoxic mechanism against colorectal cancer cells. These compounds can also suppress tumor motility, tumor aggregation, and cell invasion [7]. The limonoid compounds extracted by two different methods from *C. aurantifolia* seeds were tested for their cytotoxic activity against L5178Y lymphoma cells in vitro. The results showed that there was no difference in cytotoxic activity between the extracts, where the IC_50_ value was obtained at 8.5 g/mL for extracts extracted with supercritical CO_2_ and at 9 g/mL for extracts extracted with solvent [44].

The anticancer activity of *C. aurantifolia* against Panc-28 cells was tested by Patil et al. (2010) [114] and Patil et al. (2009) [5]. *C. aurantifolia* juice extracts using chloroform, acetone, methanol, and methanol/water (8:2) can inhibit the growth of Panc-28 cancer cells. The methanolic extract produced the highest inhibition, with an IC_50_ value of 81.20 g/mL after 72 h. Apoptosis upon cytotoxic induction was confirmed by the expression of Bax, Bcl-2, p53, and caspase-3. The mechanism of inhibition of pancreatic cancer cells is due to the activity of the compounds limonin, limonexic acid (LNA), isolemonexic acid (LCA), limonin glucoside (LG), and sitosterol glucoside (SG).

### 5.4. Insecticide

Evaluation of the insecticidal activity of *C. aurantifolia* essential oil against *Pyrrhocoris apterus* showed that at the highest concentration (100%), the insecticidal effect was up to 90%, and at the lowest concentration tested, the insecticidal effect was 10% [6]. Another study using *C. aurantifolia* essential oil was reviewed by Sarma et al. (2019) [115]. The results showed that the essential oils of the leaves and bark of *C. aurantifolia* had higher ovicidal activity (LC_50_ values of 5.26 ppm and 17.71 ppm for leaf oil and bark oil, respectively, at 72 h) compared to larvicidal activity. Essential oil from *C. aurantifolia* bark had as fast an effect as a larvicide, with an LC_50_ value of 128.81 ppm at 24 h which decreased to 106.77 ppm at 72 h, whereas essential oil from *C. aurantifolia* leaf had a slow effect, with LC_50_ values of 188.59 ppm, 107.37 ppm, and 104.59 ppm at 24 h and 48 h. Citral, as one of the compounds that make up the essential oil of *C. aurantifolia* leaves, was also tested for its ovicidal, larvicidal, and adult activities against *Aedes aegypti*. The results showed the highest ovicide activity (LC_50_ value of 4.84 ppm at 72 h), followed by larvicidal activity (LC_50_ value of 87.02 ppm at 24 h) [115].

Research on the effectiveness of *C. aurantifolia* leaf extract against the death of *A. aegypti* mosquito larvae showed that the highest larval mortality during the three-hour experiment occurred at a dose of 20%. The tendency is that the higher the dose, the higher the mortality of the larvae. This activity is associated with the toxic limonoid compounds in *C. aurantifolia* [116].

Another study was conducted to evaluate the lethal, phago deterrent, and post-embryonic development effects of the aqueous and methanolic extracts of *C. aurantifolia* leaves on third-instar larvae of *Plutella xylostella* and compare them with the synthetic pesticide Spinosad. Three concentrations of *C. aurantifolia* leaf methanolic extract (MeLE) (0.05 g/mL, 0.125 g/mL, and 0.2 g/mL), aqueous leaf extract (AqLE) (0.05 g/mL, 0.15 g/mL, and 0.2 g/mL), and synthetic pesticide (SynP) (5 g/L, 15 g/L, and 25 g/L) were tested together with solvent control against *P. xylostella* 3 instar larvae. This study revealed that *C. aurantifolia* leaf extract caused lethal, phagodetterent, and growth inhibition effects on *P. xylostella* larvae. *C. aurantifolia* can be an excellent substitute for synthetic pesticides by reducing the hazards associated with the synthetic pesticide spinosad and other pesticides. *C. aurantifolia* is considered an alternative insecticide source that may be used in dealing with *P. xylostella* using water solvents that are easily accessible to farmers [117].

### 5.5. Anti-Inflammatory

Inflammation is a reaction carried out by organisms whenever their morphological and biological constants are disturbed [118]. Several compounds that have a role as anti-inflammatory agents are polyphenols, i.e., hesperidin, naringin, and narirutin. Hesperidin is a glycoside of hesperetin. Narirutin and naringin are glycosides of naringenin. These compounds have anti-inflammatory effects in model systems and tests on humans [70].

Essential oil from the peel of *C. aurantifolia* has anti-inflammatory properties due to the presence of geranial compounds, limonene, and α-terpinene. The anti-inflammatory mechanism of this essential oil is to reduce cell migration, cytokinin production, and protein extravasation caused by carrageenan. *C. aurantifolia* essential oil can also induce myelotoxicity in mice due to its high citral content [119].

The ethanolic extract of *C. aurantifolia* bark was also reported to have strong anti-inflammatory and antinociceptive activity due to the phytoconstituents present in the extract. This shows that *C. aurantifolia* extract can be used as an effective therapeutic agent in the treatment of acute inflammation [51]. Besides that, Kasim et al. (2020) [67] proved that the ethanolic extract of *C. aurantifolia* bark affected growth mediated by the IL-6 activity of *S. typhi* bacteria in Balb/c mice. IL-6 pro-inflammatory cytokinin activity increased on day five after injection of *S. typhi* and decreased after intervention on day ten. The speed of decrease in IL-6 level was the greatest at the lime peel extract with a concentration of 750 mg/kg body weight. This shows that *C. aurantifolia* extract has the potential to be antibacterial and anti-inflammatory by reducing serum levels of IL-6, which can inhibit the growth of *S. typhi* [67]. Wardani et al. (2017) [120] also stated that the ethanolic extract of *C. aurantifolia* could improve the healing of traumatic ulcers with an optimum concentration of 25–50%, which was carried out in vitro against the *Rattus norvegicus* Wistar strain.

### 5.6. Other Bioactivities

#### 5.6.1. Antidiabetic

An antidiabetic effect of *C. aurantifolia* bark extract has been reported. In a study by Ramya et al. (2020) [121], using a methanolic extract of *C. aurantifolia* in vitro on rats, the alloxan-induced antidiabetic potential was shown. This activity is associated with the presence of secondary metabolites found in the peel of *C. aurantifolia*. Kazeem et al. (2020) [12] also showed that *C. aurantifolia* fruit could inhibit aldose reductase effectively with an IC_50_ of 138.66 g/mL. Besides that, it can also inhibit sorbitol dehydroreductase with an IC_50_ of 47.21 g/mL. Kinetic studies also show that the extract can competitively inhibit the activity of both enzymes. Enzyme inhibition through the polyol pathway is associated with the presence of flavonoids and limonoids in the *C. aurantifolia* [12].

#### 5.6.2. Leukopenia

Leukopenia is a condition with a low number of white blood cells. The active substances in *C. aurantifolia*, such as lycopene and vitamin C, can induce the proliferation of white blood cells in blood circulation. Therefore, *C. aurantifolia* has a proven therapeutic effect on leukopenia patients. Research from Ezeigwe et al. (2022) [122] also mentioned that *C. aurantifolia* juice, apart from functioning as an immune booster, can also function in weight management.

#### 5.6.3. Antiplasmodium

The antiplasmodium effect of a methanolic extract of *C. aurantifolia* leaves in vivo on Swiss albino mice was observed. Mice treated with the extract were able to live longer when compared to controls. This effect is associated with compounds such as alkaloids, flavonoids, tannins, saponins, and glycosides, which have low toxicity with LC_50_ values of 3280 mg/kg ± 0.01. This shows that *C. aurantifolia* has anti-plasmodium properties, which may be used in ethnomedicine in the treatment of malaria [123].

## 6. Microencapsulation in Food Application

Functional food development faces several challenges, especially regarding the direct use of bioactive ingredients. These bioactive components plausibly could experience deactivation caused by their stability against the environment or reactive with the food matrix, or degradation thus changing the tastes and odors. Microencapsulation is available as an approach that can solve these problems. Additionally, microencapsulation can provide a controlled or targeted delivery [124]. The success of targeted microencapsulation can be measured based on the behavior of bioactive ingredients during processing, storage, and after consumption [17].

Various ways of encapsulation have already been applied in the food industry to improve the taste, texture, nutrient bioavailability, and shelf life of foods—for instance, in the cereal, beverage, bakery, dairy [125], meat, and fruit product industries and their derivatives, with good results [18]. The microencapsulation method could also combine the bioactive components and functionalization of the food matrix. Several microencapsulation techniques such as fluidized bed coating, spray cooling, spray drying, extrusion, and coacervation were already commonly used [126]. Such a simple method as coacervation was used to encapsulate bioactive components of the lime peel oils using ethyl cellulose coatings [127] and complex coacervation methods using alginate gelatin coatings [128]. Moreover, another microencapsulation of lime peel extract and its juice using gum arabic and dextrin coatings has also been studied for tea bag applications.

However, some obstacles that may be faced in validating the quality and stability of microencapsulated materials are cost, encapsulation efficiency, release rate, water solubility, particle size, and taste [18]. It is important to pay attention to the materials chosen for microencapsulation, especially when it involves food applications, such as coatings and crosslinkers. The body’s safety and the safety of these material selections are priorities. It is best to stay away from potentially harmful ingredients such as hexamethylenediisocyanate [129], carbodiimide [130], and glutaraldehyde [131]. Biopolymers materials commonly used in food applications can be seen in Figure 8.

## 7. Conclusions

The herbal plant *C. aurantifolia* has beneficial bioactive ingredients. Using various extraction techniques, the bioactive substances can be drawn out of *C. aurantifolia* plant parts such as the fruit, peel, leaves, seeds, roots, stems, and bark. The review reveals that the composition and quantity of compounds vary depending on the part of *C. aurantifolia* as well as the extraction method. Further investigation of their extract and essential oil has shown their potential to be developed as anticancer, antioxidant, anti-inflammatory—and other—drugs. Bioactive components are generally prone to degradation, either during storage or food processing; thus, microencapsulation can be used as an alternative method to solve the problems.

Finding appropriate extraction techniques for the bioactive components found in *C. aurantifolia* plants and their packaging method might be the successful key for their further application in the related industry. Therefore, it is anticipated that the data presented will serve as the foundation for future research on the development of the use of *C. aurantifolia*, particularly in food, chemical, pharmaceutical, and other industries.

## Figures and Tables

**Figure 1 foods-12-02036-f001:**
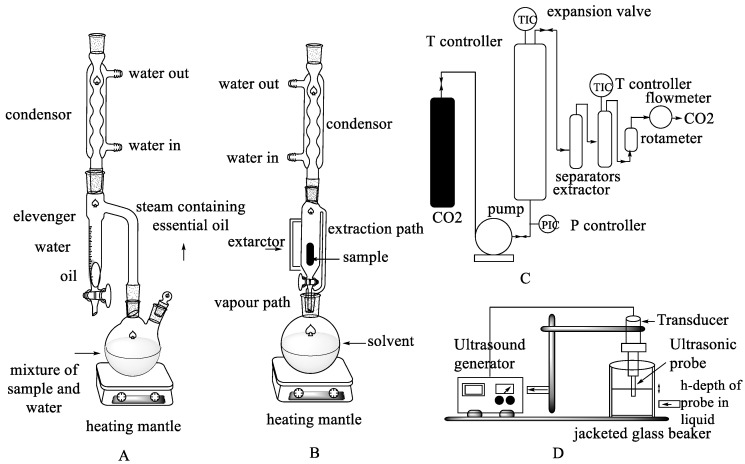
Schematic of the secondary metabolite extraction method. (**A**) Hydrodistillation (**B**) Soxhlation (**C**) Supercritical-CO_2_ extraction (**D**) Ultrasound-Assisted Extraction (UAE).

**Figure 2 foods-12-02036-f002:**
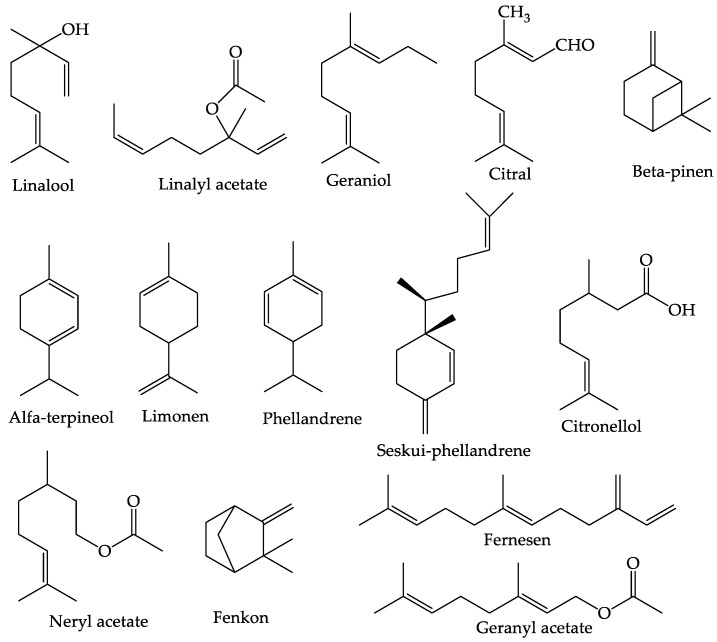
Structure of the main components in the essential oil of *C. aurantifolia*.

**Figure 3 foods-12-02036-f003:**
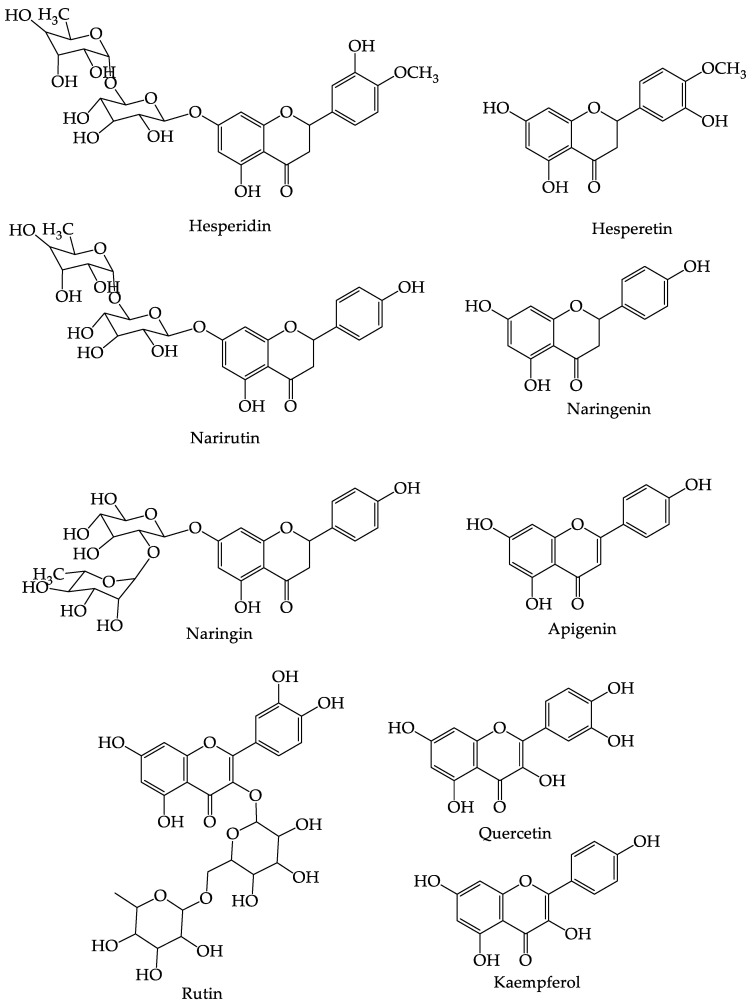
The structure of the flavonoid compounds found in *C. aurantifolia*.

**Figure 4 foods-12-02036-f004:**
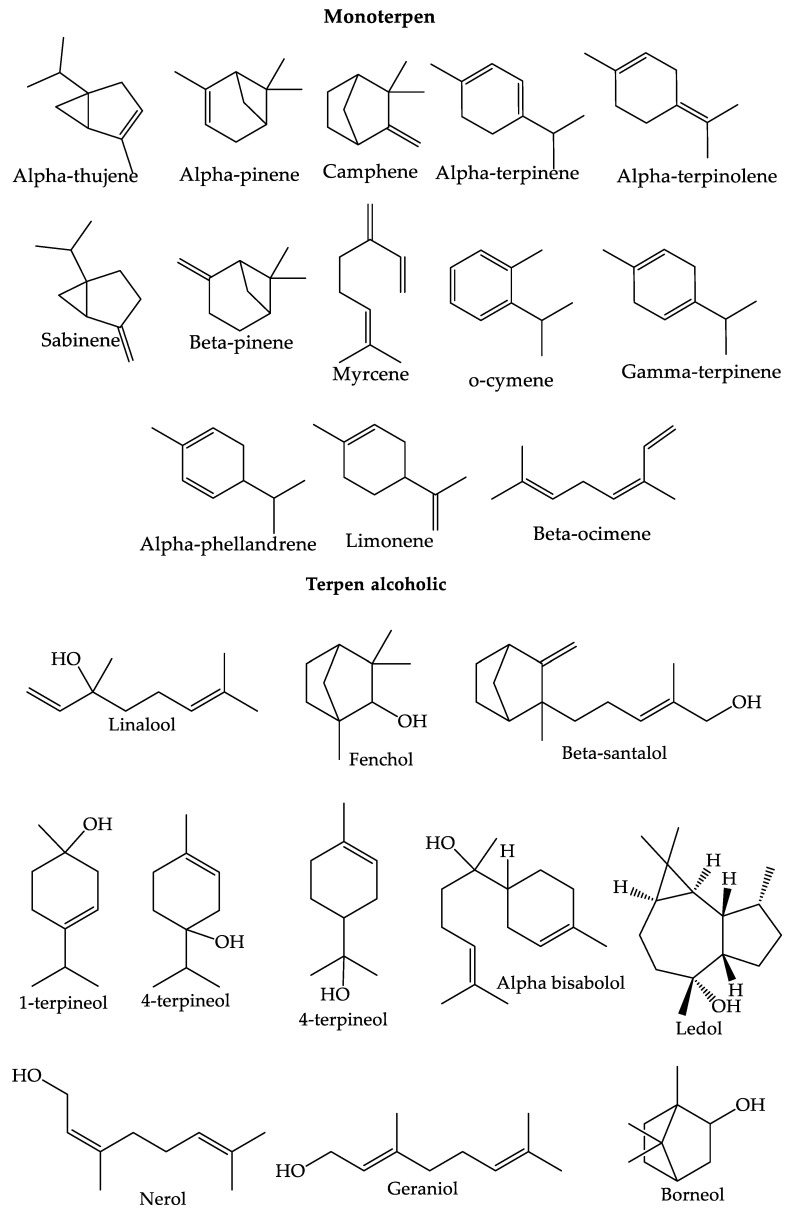
The structure of the terpenoid compounds found in *C. aurantifolia*.

**Figure 5 foods-12-02036-f005:**
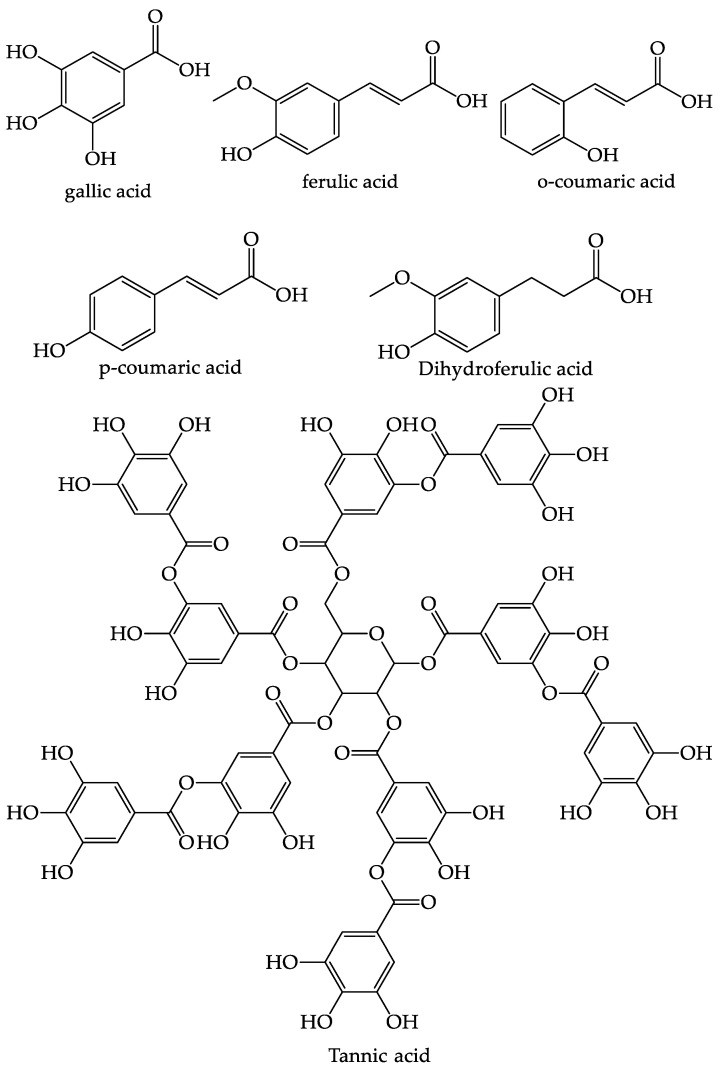
The structure of the phenolic compounds found in *C. aurantifolia*.

**Figure 6 foods-12-02036-f006:**
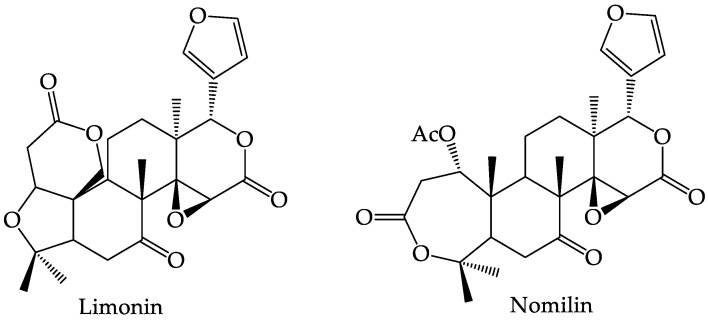
Structure of Limonin and Nomilin.

**Figure 7 foods-12-02036-f007:**
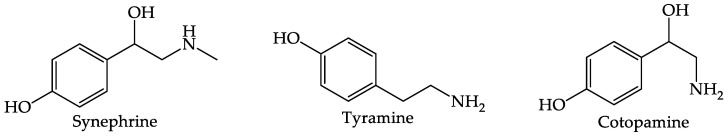
The structure of alkaloid compounds found in *C. aurantifolia*.

**Figure 8 foods-12-02036-f008:**
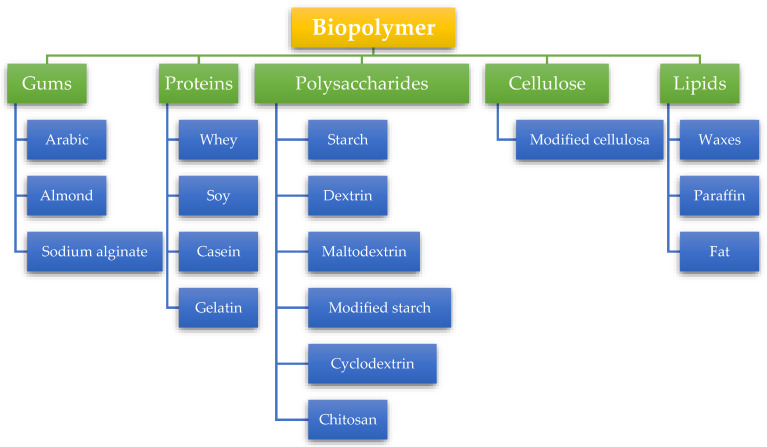
Biopolymers materials commonly used in food applications.

**Table 1 foods-12-02036-t001:** Comparison of extraction methods on *C. aurantifolia*.

No	Secondary Metabolite	Technique	Plant Parts	Yields	Reference
1	Essential Oil	Hydrodistillation	milled peel whole peels leaf	5.45% 0.97% 0.75%	[38] [38] [39]
		Supercritical-CO_2_ extraction	milled peel whole peels	7.93% 1.98%	[38]
		Steam distillation	peel leaf	1.5% 0.75%	[39]
		Maceration	peel	*n*-hexane = 5.24% distilled water = 1.67% ethanol = 4.20%	[40]
		Soxhlation	peel	*n*-hexane = 6.15% ethanol = 4.89%	[40]
		Cold pressed	leaf	0.5%	[6,39]
2	Flavonoids	Maceration	fruit	Ethanol = 7.83 ± 2.66 mgCA/g extract Methanol 70% = 6.27 ± 0.39 mgCA/g extract Aquades = 6.52 ± 0.77 mgCA/g extract	[41]
		Maceration with 95% ethanol	leaf	Leaf of Nakhal 64.2 ± 2.8 (μg of QE/mg dry extract) Leaves from Nizwa 41.38 ± 5.5 (μg of QE/mg dry extract)	[11]
3	Phenolic	Ultrasound Assisted Extraction (UAE),	peel	Yields increased when compared maceration extract	[42]
		Low Power Ultrasound-Assisted Extraction	peel	3083.61 mg gallic acid equivalent (eq) 100 g^−1^ dry weight of the total phenolic component	[43]
4	Limonoids	Solvent extraction, supercritical-CO_2_ extraction	seed	11.39% dry base ± 1.3 for methanol 10.37% dry base ± 2.21 for acetone	[44]

**Table 2 foods-12-02036-t002:** Bioactive components found in *C. aurantifolia* plant parts.

Bioactive Components	Section *C. aurantifolia*							Reference
	Leaf	Fruit	Rind	Seed	Stem	Root	Bark	
Essential oil	√	-	√	-	-	-	-	[2,11,48,49,50,51,52,53]
Flavonoids	√	√	√	√	√	√	√	
Terpenoids	√	-	√	√	-	√	√	
Phenolic	√	√	√	√	√	-	√	
Limonoids	-	-	-	√	-	-	-	
Alkaloids	√	√	√	√	√	√	√	

**Table 3 foods-12-02036-t003:** Comparison of total phenolic content in *C. aurantifolia* plant parts.

Leaf	Fruit	Peel	Seed	Reference
4.675 mg/g wet weight	2.243 mg/g wet weight	6.954 mg/g wet weight	19.87 ± 0.03 mg/100 g wet weight	[53,65]

**Table 4 foods-12-02036-t004:** Antibacterial activity of plant parts of *C. aurantifolia*.

No	Part	Extraction Solvent	Bacteria	Method	Reference
1	Essential Oil	-	*Azotobacter chroococcum, Serratia marcescens, Priestia megaterium*	Disk diffusion method	[6]
			*Micrococcus luterus*		
			*Staphylococcus aureus, Staphylococcus epidermidis, E. coli, Klabsiella pneumoniae*		[91]
2	Peel	Hexane	*Mycobacterium tuberculosis* H37rv	Microplate Alamar Blue Assay (MABA)	[92]
		Ethanol 48%, 72%, 96% Ethyl acetate	*S. aureus* ATCC 25923 and *E. coli* ATCC 25922	Disk diffusion method	[93]
		Ethanol	*Bacillus cereus, E. Coli*	Disk diffusion method	[71]
3	Stem	Ethanol, Aquades	*S. aureus, Pseudomonas aeruginosa, Pseudomonas mirabilis* and *K. Pneumoniae*	Disk diffusion method	[94]
			*E. coli, Bacillus megaterium, P. aeruginosa, Enterobacter aerogenes, Salmonella* spp., *Proteus myxofaciens, K. pneumoniae, Kluyvera ascorbata, S. aureus*	Dilution	
4	Leaf		*S. aureus*	Disk diffusion method	[76]
		Higher in ethanol than distilled water	*Shigella, Klebsiella, E. coli, S. typhi*	Disk diffusion method	[89]
		Hydroalcoholic	*S. aureus, E. coli, K. pneumoniae, Pseudomonas* spp.	Disk diffusion method	[95]
		Aquades, Ethanol, Methanol, Acetone	*S. aureus, E. coli, K. pneumoniae, P. aeruginosa, Aspergillus niger, Mucor mucedo, Penicillium notatum, Candida albicans*		[49]
			*E. coli* ATCC 25922	The Kirby–Bauer	[96]
		Aqueous, acetone, ethanol	*E. coli, Pseudomonas, S. aureus, Klebsiella*	Disk diffusion method	[97]
5	Fruit	Aquades, Ethanol, Tuak, Seamann’s Schnapps, Fermented Water from 3 days Soaking Corn Powder/Corn Paste (Ekan-Ogi/Omi-Ogi)	Gram-negative *Serratia* spp., *Salmonella paratyphi,* *Shigella flexnerri, P. aeruginosa, K. pneumoniae, Citrobacter* spp., and *E. coli* Gram-positive (*S. aureus, Enterococcus feacalis*) fungi (*Aspergillus niger, Candida albicans*) three anaerobic bacteria (*Clostridium* spp., *Bacteroides* spp. and *Porphyromonas* spp.)	Disk diffusion method	[98]
6	Juice	Aquades	*E. coli*	Disk diffusion method	[99]
7	Seed	Ethanol, Chloroform, Methanol	*Bacillus subtilis* NCTC 8236. *S. aureus* ATCC 25923. *E. coli* ATCC 25922. Proteus vulgaris ATCC 6380. *Klebsiellas* pp. ATCC 53657. *Shigella* spp. NCTC 4837.	Disk diffusion method	[2]
8	*Crude*		*E. coli*	Kirby–Bauer susceptibility test method	[100]

**Table 5 foods-12-02036-t005:** Comparison of antioxidant activity on plant parts of *C. aurantifolia*.

No	Source	Method	Antioxidant Activity	Reference
1	Essential oil	DPPH	74.5 ± 0.5%, with a corresponding 442 ± 2.3 TEAC	[6]
		DPPH	IC_50_ 2.19 mg/mL	[54]
		DPPH	IC_50_ 12.85 µL/mL, ascorbic acid 5.28 µL/mL	[104]
		DPPH	IC_50_ *C. aurantifolia* var. Bearss 4.32 mg/L, var. Mexican 1.62 mg/L and var. sans epines 0.26 mg/L	[105]
		ABTS	IC_50_ 2.00 mg/mL	[54]
2	Juice	FRAP	4.98 µmol Fe(II)/g (in immature *C. aurantifolia*)	[74]
		DPPH	Ripe and unripe *C. aurantifolia* juices showed 91.02% and 96.14% scavenging against DPPH at a concentration of 100 µL	[102]
3	Fruit juices and peel	Low-density lipoprotein (LDL)	10 µL of juice inhibited LDL oxidation and increased with increasing concentration	[106]
4	Leaves and bark (methanolic extract)	DPPH & FRAP	Reducing ability ranges from 112.1–146.0 µmol L^−1^ Fe(II) g^−1^ IC_50_ ranges from 91.4–107.4 µgmL^−1^	[14]
5	Leaves (ethanolic extract)	DPPH	Both extracts of lime leaves from Nakhal and Nizwa showed moderate antioxidant activity depending on the concentration range (11.79–56.89 and 10.11–51.91%)	[11]
		DPPH & FRAP	IC_50_ = 83.89 ppm 70% ethanolic extract and IC_50_ = 88.02 ppm 96% ethanolic extract FRAP test 188.74 mg AaE/g 96% ethanolic extract and 181.034 mg AaE/g 70% ethanolic extract.	[4]
6	Fruit (methanolic extract)	DPPH	IC_50_ = 1793.06 g/mL	[107]
7	Peel	DPPH	The ethyl acetate fraction has the largest IC_50_ which is 457.6 ppm	[108]

## Data Availability

Data is contained within the article.

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
