# Peer review of "Bioactive Components and Their Activities from Different Parts of *Citrus aurantifolia* (Christm.) Swingle for Food Development"

_foods, 2023, doi:10.3390/foods12102036_

Round 1
Reviewer 1 Report
This manuscript aimed to discuss the active components in different parts of C. aurantifolia and their biological activities. Microencapsulation methods to protect bioactive components in food are investigated. The extraction methods and their yield from the plant matrix are reported. Therefore, I suggest the author complements the content, articulates the obscure content, and corrects the format.
1. What is the main point of the article and what is its novelty?
2. The abstract is not sufficiently condensed. And please add a sentence which shows the necessity of the study.
3. The logic of the introduction is not very fluid. For example, should it be briefly introduced first on Page 2. Line 51. What is the microencapsulation method. There are many ways to prevent the degradation of active ingredients. What are the differences between the various methods? And why only discussed the microcapsule?
4. Title, figure notes and table format issues were not uniform. For example, Page 3, Line 109 (a) is omitted, and Figure C and D in Figure 1 are not aligned. All compound names that appear in figures are consistently and clearly arranged. Tables 4 and 5 need bold headings. The above content needs to be carefully modified and checked.
5. In addition to the impact on the yield between different extraction methods, whether there is also impact on the biological activity, it can be done for further in-depth discussion. And the advantages and disadvantages of different extraction methods can be discussed further.
6. Although you comprehensively summarize the content, extraction methods, distribution and biological activities of the main active ingredients in C. aurantifolia, you need to express your own opinions on this basis. For example, how to use these active ingredients reasonably to play a physiological role, or whether there are any aspects that need in-depth study, etc.
7. The logical structure of some chapters is unclear due to the absence of introductions and summaries at the beginning and end of each paragraph. In addition, it is necessary to establish a link between paragraphs and paragraphs. And C. aurantifolia’s anti-cancer activity use form to show will be better.
8. Is the application of microencapsulation of C. aurantifolia is supported by relevant patents or literature? If you can find the relevant content, please add it in the text.
9. The conclusion should contain more, as it is more of an afterthought. The conclusion will not only summarize the content explained before, but also the shortcomings of the previous research and appropriate solutions can be proposed. Or suggest the direction and outlook for future development.
This manuscript aimed to discuss the active components in different parts of C. aurantifolia and their biological activities. Microencapsulation methods to protect bioactive components in food are investigated. The extraction methods and their yield from the plant matrix are reported. Therefore, I suggest it be accepted after the author complements the content, articulates the obscure content, and corrects the format.
1. What is the main point of the article and what is its novelty?
2. The abstract is not sufficiently condensed. And please add a sentence which shows the necessity of the study.
3. The logic of the introduction is not very fluid. For example, should it be briefly introduced first on Page 2. Line 51. What is the microencapsulation method. There are many ways to prevent the degradation of active ingredients. What are the differences between the various methods? And why only discussed the microcapsule?
4. Title, figure notes and table format issues were not uniform. For example, Page 3, Line 109 (a) is omitted, and Figure C and D in Figure 1 are not aligned. All compound names that appear in figures are consistently and clearly arranged. Tables 4 and 5 need bold headings. The above content needs to be carefully modified and checked.
5. In addition to the impact on the yield between different extraction methods, whether there is also impact on the biological activity, it can be done for further in-depth discussion. And the advantages and disadvantages of different extraction methods can be discussed further.
6. Although you comprehensively summarize the content, extraction methods, distribution and biological activities of the main active ingredients in C. aurantifolia, you need to express your own opinions on this basis. For example, how to use these active ingredients reasonably to play a physiological role, or whether there are any aspects that need in-depth study, etc.
7. The logical structure of some chapters is unclear due to the absence of introductions and summaries at the beginning and end of each paragraph. In addition, it is necessary to establish a link between paragraphs and paragraphs. And C. aurantifolia’s anti-cancer activity use form to show will be better.
8. Is the application of microencapsulation of C. aurantifolia is supported by relevant patents or literature? If you can find the relevant content, please add it in the text.
9. The conclusion should contain more, as it is more of an afterthought. The conclusion will not only summarize the content explained before, but also the shortcomings of the previous research and appropriate solutions can be proposed. Or suggest the direction and outlook for future development.
Author Response
- What is the main point of the article and what is its novelty?
There is currently not a review of the chemical make-up of different plant sections, biological activity, or application of C. aurantifolia as food-grade microcapsules. Therefore,this review article discusses the bioactivity of compounds from C. aurantifolia plant parts, including leaves, fruit, peels, roots, and seeds, as well as their bioactivity as anticancer, antioxidant, anti-inflammatory, and other bioactivities to maximize health benefits
- The abstract is not sufficiently condensed. And please add a sentence which shows the necessity of the study.
we add a sentence about the importance of further research in the conclusion. “It is a challenge for researchers to use the right extraction techniques to more thoroughly investigate the bioactive components found in C. aurantifolia plants. The development of their use is a problem as well, particularly in the food industry and other industries that use antibacterial agents, antioxidants, and other bioactivities to enhance food quality, shelf life, and nutrition.”
- The logic of the introduction is not very fluid. For example, should it be briefly introduced first on Page 2. Line 51. What is the microencapsulation method. There are many ways to prevent the degradation of active ingredients. What are the differences between the various methods? And why only discussed the microcapsule?
we add some introduction regarding microencapsulation and the advantages of microencapsulation when applied to protect bioactive components. because of the focus on microencapsulation so it does not discuss other methods further
- Title, figure notes and table format issues were not uniform. For example, Page 3, Line 109 (a) is omitted, and Figure C and D in Figure 1 are not aligned. All compound names that appear in figures are consistently and clearly arranged. Tables 4 and 5 need bold headings. The above content needs to be carefully modified and checked.
we have reviewed and corrected the writing of image and table titles so that they are uniform
- In addition to the impact on the yield between different extraction methods, whether there is also impact on the biological activity, it can be done for further in-depth discussion. And the advantages and disadvantages of different extraction methods can be discussed further.
we add a few sentences about extraction and discuss further about a number of things such as the advantages and disadvantages of the extraction method. Meanwhile, the effect of the extraction method on biological activity is still rarely studied, so we will not discuss it in this review article
- The logical structure of some chapters is unclear due to the absence of introductions and summaries at the beginning and end of each paragraph. In addition, it is necessary to establish a link between paragraphs and paragraphs. And aurantifolia’santi-cancer activity use form to show will be better.
we added some introduction and conclusion in some parts
- Is the application of microencapsulation of aurantifoliais supported by relevant patents or literature? If you can find the relevant content, please add it in the text.
we add some research on the microencapsulation of citrus aurantifolia
- The conclusion should contain more, as it is more of an afterthought. The conclusion will not only summarize the content explained before, but also the shortcomings of the previous research and appropriate solutions can be proposed. Or suggest the direction and outlook for future development
we correct the conclusions and add some future views regarding research on the development of citrus aurantifolia

Reviewer 2 Report
The manuscript titled “Bioactive Components and Their Activities from Different Parts of Citrus aurantifolia (Christm.) Swingle for Food Development” described various bioactive components and its therapeutic potential. In addition, authors discus microencapsulation to prevent the bioactive component deteriorations from various potential threats in terms of oxidative stability or others. The article is been written well, however it needs major revision in further submission. Reviewer comments has been marked in pdf file, kindly refer to pdf pages.

Round 2
Reviewer 2 Report
The manuscript titled “Bioactive Components and Their Activities from Different Parts of Citrus aurantifolia (Christm.) Swingle for Food Development” has been revised as per the comment raised by the reviewer. However, I feel that author needs to update whole manuscript concerning English and grammar.
The manuscript titled “Bioactive Components and Their Activities from Different Parts of Citrus aurantifolia (Christm.) Swingle for Food Development” has been revised as per the comment raised by the reviewer. However, I feel that author needs to update whole manuscript concerning English and grammar.
Author Response
author needs to update whole manuscript concerning English and grammar.
respons: we have reviewed and tried to improve english and grammar